# New Viral Sequences Identified in the Flavescence Dorée Phytoplasma Vector *Scaphoideus titanus*

**DOI:** 10.3390/v12030287

**Published:** 2020-03-06

**Authors:** Sara Ottati, Marco Chiapello, Luciana Galetto, Domenico Bosco, Cristina Marzachì, Simona Abbà

**Affiliations:** 1Dipartimento di Scienze Agrarie, Forestali ed Alimentari, DISAFA, Università degli Studi di Torino, Largo Paolo Braccini 2, I-10095 Grugliasco, Italy; sara.ottati@unito.it (S.O.); domenico.bosco@unito.it (D.B.); 2Consiglio Nazionale delle Ricerche—Istituto per la Protezione Sostenibile delle Piante (CNR-IPSP), Strada delle Cacce 73, I-10135 Torino, Italy; marco.chiapello@ipsp.cnr.it (M.C.); luciana.galetto@ipsp.cnr.it (L.G.); cristina.marzachi@ipsp.cnr.it (C.M.)

**Keywords:** Flavescence dorée, *Scaphoideus titanus*, insect viruses, metatranscriptomics, *Vitis vinifera*

## Abstract

**(1) Background:** The leafhopper Scaphoideus titanus is the primary vector of Flavescence dorée phytoplasma (FDp) in European vineyards. Flavescence dorée is one of the most severely damaging diseases of Vitis vinifera and, consequently, a major threat to grape and wine production in several European countries. Control measures are compulsory, but they mainly involve large-scale insecticide treatments, with detrimental impacts on the environment. One possible solution is to exploit the largely unexplored genetic diversity of viruses infecting S. titanus as highly specific and environmentally benign tools for biological control. **(2) Methods:** A metatranscriptomic approach was adopted to identify viruses that may infect individuals caught in the wild in both its native (United States) and invasive (Europe) areas. Reverse transcription PCR was used to confirm their presence in RNA pools and explore their prevalence. **(3) Results:** We described nine new RNA viruses, including members of “Picorna-Calici”, “Permutotetra”, “Bunya-Arena”, “Reo”, “Partiti-Picobirna”, “Luteo-Sobemo” and “Toti-Chryso” clades. A marked difference in the diversity and abundance of the viral species was observed between the USA population and the European ones. **(4) Conclusions:** This work represents the first survey to assess the viral community of a phytoplasma insect vector. The possibility to exploit these naturally occurring viruses as specific and targeted biocontrol agents of *S. titanus* could be the answer to increasing demand for a more sustainable viticulture.

## 1. Introduction

*Scaphoideus titanus* Ball. (Hemiptera: Cicadellidae) is a phloem-feeding nearctic leafhopper that was probably introduced accidentally from North America to Europe in the late 19^th^ century, in the attempt to find a solution against downy mildew and phylloxera epidemics occurring in Europe at that time [1]. Genetic characterization of American and European *S. titanus* populations revealed that the latters are likely to descend from a single introduction from the United States [2]. In its original range, *S. titanus* was mostly captured in open woodlands, on wild *Vitis* species and only rarely on cultivated grapevines [3,4]. In Europe, by contrast, it has been reported mainly on *Vitis vinifera*, although it is known to thrive in abandoned vineyards and wild grapevines [5,6,7,8]. *S. titanus* is the primary vector of Flavescence dorée phytoplasma (FDp) in vineyards, and it has so far, spread to many European grape-growing countries, such as Italy [9], France [10], Hungary [11] and Switzerland [12]. Phytoplasmas are obligate bacterial pathogens, confined mainly in the phloem tissue. Upon feeding on infected plants, sap-sucking insects, such as *S. titanus*, can passively acquire them and transmit the pathogen to healthy plants [13]. FDp is a quarantine pest of grapevine in the European Union. It typically causes color aberrations and downward-rolled margins in foliage, lack of lignification in shoots, reduced fruit setting, flower withering, bunch wilting and berry drop. In the end, infected plants either die or recover from the disease, but with a significant reduction in productivity [14]. The overall economy of wine-producing regions is not only affected by yield losses caused by this disease, but also by the expensive compulsory containment measures [15,16,17]. The only currently available control strategies focus on the reduction of vector populations in vineyards, and include uprooting of symptomatic grapevines, compulsory insecticide applications with specific timing, removal of wild vines that may act as refuges for the insect, and hot water treatments of rootstocks, scions or grafted cuttings to kill deposited eggs [18]. At the same time, concerns are mounting especially over the effects of pesticides on the environment, non-target organisms and human health. Consequently, it is essential to develop more sustainable alternatives to the use of chemicals. A possible option would be to exploit the largely unexplored genetic diversity of viruses infecting *S. titanus* for developing highly specific and environmentally benign tools for biological control.

Baculoviruses have been successfully used as safe and selective bioinsecticides to control mainly Lepidoptera, since their first commercial use in 1975 [19]. Nevertheless, the use of baculoviruses is still limited by their narrow host range, which exclude major plant pests such as those belonging to the order Hemiptera. Only two members of the families *Parvoviridae* and *Nudiviridae* were used so far for biological control of insect pests other than Lepidoptera: Cockroach densonucleosis virus against *Periplaneta fuliginosa* (Blattodea) [20] and Oryctes virus against *Oryctes rhinoceros* (Coleoptera) [21].

In this work we applied RNA-Seq to describe putative viruses that may infect wild-caught *S. titanus* populations in both its invasive (Europe) and native (United States) geographical areas. Comparisons between the two sampling areas allowed for the description of two completely different viral profiles, characterized by a reduction in the number of potential enemies in the European populations. Nine new complete or near-complete viral genomes were identified, representing seven virus families. An iflavirus identified in Europe, in particular, represented a potential candidate for future isolation and pathogenicity assays or for the construction of virus-induced gene silencing (VIGS) vectors to transiently knock-down insect genes involved in phytoplasma acquisition/transmission.

## 2. Materials and Methods

### 2.1. Insect Collection

*S. titanus* specimens used for RNA-seq were collected from the field with a sweeping net in 15 European sites and in one USA site during summer 2018 (Appendix A). Sampling locations, GPS coordinates in decimal degrees, collection dates and number of sampled insects are reported in Table 1. All insects, except those collected in Piedmont, were immediately stored and shipped in RNA*later* (Thermo Fisher Scientific Inc., Waltham, MA, USA).

After the virome assembly, four specimens were used for verifying the possible integration of some viral sequences into the insect genome. They were collected from the field with yellow sticky traps left for two days in highly infested vineyards (Montà and Dogliani, Piedmont, Italy) (Table 1) localized in the same geographical area from which insects were sampled to construct one of the two Italian libraries for RNA sequencing.

### 2.2. DNA Extraction, RNA Extraction, Library Preparation and Sequencing

Adult insects were frozen with liquid nitrogen, crushed with a micropestle in sterile tubes, and then subjected to either DNA or RNA extraction. DNA was extracted according to the method described by Marzachì et al. [22]. For RNA extraction, samples were homogenized in TRI Reagent (Zymo Research, Irvine, CA, USA) and centrifuged for 1 min at 12,000 g at 4°C to remove cell debris. RNA was extracted from supernatants with Direct-zol RNA Mini Prep kit (Zymo Research, Irvine, CA, USA), following the manufacturer’s protocol with DNAse treatment. Concentration, purity, and quality of DNA and RNA extractions were estimated using a Nanodrop spectrophotometer.

RNA extracts were pooled for library construction according to the country of origin of the insects: St_FR library for France, St_CH library for Switzerland, St_HU library for Hungary, and St_USA for United States of America (Table 1, Appendix A). The Italian samples were grouped in two libraries: specimens collected in Piedmont, Lombardy and Veneto were pooled to construct the St_IT1 library, whereas the remaining insects from Friuli-Venezia Giulia and Abruzzo were pooled in the St_IT2 library (Table 1, Appendix A).

Library preparation and sequencing were conducted by Macrogen Inc. (Seoul, South Korea). The Ribo-Zero Gold Kit (Human/Mouse/Rat) (Illumina Inc., San Diego, CA, USA) was used to deplete ribosomal RNA. cDNA libraries were prepared using a TruSeq Stranded Total RNA Library Prep Kit (v2) (Illumina Inc. San Diego, CA, USA) and were sequenced from both ends (100 bp) using the NovaSeq System.

### 2.3. Bioinformatic Analysis

The pre-assembly steps were performed using the suite of bioinformatic tools called BBTools v38.70 [23]. Raw paired-end files were processed for removal of Illumina adaptor sequences and artifacts, and for length and ribosomal filtering via kmer matching using BBDuk. To avoid unwanted environmental contaminants, the remaining reads were further analyzed to remove reads matching to human, mouse and dog sequences, using BBMap. Finally, BBMerge and BBNorm were used for merging overlapping paired reads and normalizing read coverage before assembling with Trinity v2.6.6 [24]. Some of the resulting sequences were further assembled by CAP3 v3 (overlap length cutoff = 65; overlap percent identity cutoff N = 90) [25]. DIAMOND v0.9.24.125 [26] with an E-value cut-off of 0.0001 was used to perform a blastx search for matches to viral genomes in the NCBI non redundant protein database (version October 2018). Putative viral sequences were further checked and analyzed manually, in an effort to exclude as much as possible viruses associated with insect food (especially grapevine), gut microflora or possible parasites. Such filtering was based on similarities with known plant, fungi and nematode viruses, and on the number of reads mapping on each viral transcript, assuming that the more common the virus was, the more likely it was infecting *S. titanus*. Bowtie2 [27] was used to map reads against the reference transcripts with parameters that selected only reads that mapped in pairs with the right orientation (--no-mixed –no-discordant). Reads mapping onto viral genomes were expressed as RPKM (Reads Per Kilobase of transcript per Million mapped reads).

The bioinformatic command lines and parameters used in RNA-seq data analyses are provided as Appendix A.

Viral sequences have been deposited in the GenBank database under the accession numbers MN982379-MN982405.

### 2.4. Phylogenetic Trees

Phylogenetic relationships were inferred on the basis of putative RNA-dependent RNA polymerases (RdRps), as they shared conserved domains across all RNA viruses. All the phylogenies described in this manuscript used clades, nomenclature and sequences provided by Shi et al. [28]. RdRp amino acid sequences of the newly discovered viruses were trimmed to include only conserved domains and aligned using MAFFT v7 [29] with the E-INS-i algorithm. Phylogenetic trees were then generated using the maximum likelihood approach (ML) implemented in IQ-TREE [30] with default parameters.

### 2.5. PCR and RT-PCR

In order to independently confirm the RNA-seq results, reverse transcriptase PCR (RT-PCR) was used to obtain short amplicons from at least one segment (in the case of multipartite viruses) of the newly identified viruses.

For each RNA pool used for library construction, cDNA was synthesized from total RNA (200 ng) using the High Capacity cDNA reverse transcription kit (Thermo Fisher Scientific Inc., Waltham, MA, USA). Specific primers were designed for all the newly identified viruses. RT-PCR was also conducted to determine the prevalence of the European viruses in the original unpooled RNA samples. The absence of contaminating genomic DNA was verified, including in the PCR step samples without the reverse transcription step.

Additional primers were designed on putative endogenous viral elements (EVEs) and used in PCR (without reverse transcriptions) on *S. titanus* genomic DNA. All primer sequences and amplification conditions used in this work are listed in Appendix A. The resulting amplicons were validated by Sanger sequencing at BMR Genomics (Padua, Italy).

## 3. Results

We characterized the total transcriptome of six *S. titanus* pools: five from European locations (St_IT1, St_IT2, St_HU, St_FR and St_CH) and one from a USA location (St_USA). RNA sequencing resulted in 110 million to 130 million reads per library, which were independently assembled into contigs and compared to NCBI protein non redundant database for virus discovery. These transcriptome data allowed us to identify nine new viral RNA genomes associated to *S. titanus*: two in the European libraries and seven in the USA one (Table 2). Five of the viruses described here are expected to have a positive-sense single-stranded RNA (+ssRNA) genome, one a negative-sense single-stranded RNA (−ssRNA) genome and three a double-stranded RNA (dsRNA) genome. Only complete or near complete viral genomes were considered in this computation, i.e., genomes comprising the complete coding potential that should be expected from the existing virus taxa they are most closely related to. None of these genomes featured premature stop codons or repeat sequences, so they unlikely represented endogenous viral elements (EVEs), namely integrations of DNA and nonretroviral RNA viruses into the host genome. According to these strict criteria, the St_HU library was the only one in which we could not identify any sequence matching to viruses deposited into public databases.

The presence of viral genomes in the original RNA samples was confirmed by RT-PCR coupled with Sanger sequencing. The newly identified viruses were tentatively named according to their host (*S. titanus*) and phylogenetic relationships.

### 3.1. Single-Stranded Positive-Sense RNA Viruses

Three contigs, two from St_IT1 and St_FR libraries and one from the St_USA library, showed the highest similarity (83% amino acid identity, 51% query coverage) with the polyprotein of Graminella nigrifrons virus 1, which was assigned to the family *Iflaviridae* (Table 2).

According to the International Committee on Taxonomy of Viruses (ICTV) species demarcation criteria for the genus Iflavirus [31], isolates/strains belong to the same iflavirus species if the sequence identity at the amino acid level, between their coat proteins, is above 90%. On this basis, the two European genomes significantly differed from the known iflavirus sequences, but not from each other, so they were likely to represent a single new iflavirus species, hereafter named Scaphoideus titanus iflavirus 1. The USA iflavirus showed a percentage of amino acid identity below 90% if compared to other known iflaviruses and Scaphoideus titanus iflavirus 1. Therefore, it could represent another new iflavirus species, from here on named Scaphoideus titanus iflavirus 2.

The genome of iflaviruses is a monopartite +ssRNA genome, which encodes a large polyprotein auto-catalytically cleaved into structural and nonstructural component peptides: starting from the N-terminus, capsid proteins, followed by three non-structural proteins, an RNA helicase, a 3C-like cysteine protease, and an RdRp. A Pfam search revealed the presence of these peptides in both Scaphoideus titanus iflaviruses (Appendix A). The two new RdRps were aligned to those of viruses within the “Iflaviridae-Secoviridae related cluster” [28] with the addition of the first three best hits retrieved from the blastx analysis (GenBank accessions: APD68841.1, YP_009553259.1 and YP_009129265.1). The phylogenetic analysis confirmed that both sequences were closely related to iflaviruses identified from other hosts of the family Cicadellidae (*Graminella nigrifrons*, *Psammotettix alienus* and *Euscelidius variegatus*) (Figure 1A, Appendix A).

Four contigs from the St_USA library were tentatively assigned to two sobemo-like viruses, hereafter referred to as Scaphoideus titanus sobemo-like virus 1 and 2 (Table 2). The two RdRps showed the highest similarity with those of Hubei sobemo-like virus 26 and Hubei sobemo-like virus 24, respectively. Despite the fact that most sobemo-like viruses described by Shi et al. [28] have a +ssRNA monopartite genome, we could reasonably hypothesize that each sobemo-like genome identified in *S. titanus* has a bipartite genome: the longer fragment (around 3000 bp) encodes two non-overlapping ORFs coding for a trypsin-like serine protease and an RdRp; the second shorter genomic fragment (around 1500 bp) codes for a putative capsid protein (Appendix A). Such hypothesis was based on similar depth of coverage across the viral segments, and, above all, on the presence of the same heptanucleotide at the termini of the two segments assigned to Scaphoideus titanus sobemo-like virus 1. Consequently, the other two sequences most likely belonged to Scaphoideus titanus sobemo-like virus 2. The correct two-by-two association between long and short segments was also confirmed by RT-PCR: one insect was positive only for one couple of segments and another insect was positive only for the other couple (data not shown). Notably, both hypothetical capsid proteins were similar to a hypothetical protein identified in a nodavirus (Naganuma virus). The two newly identified sobemo-like viruses appeared to belong to two different lineages within the large clade that grouped viruses identified from a variety of arthropods (Figure 1B, Appendix A).

A contig from the St_USA library showed the highest similarity (39% amino acid identity, 39% query coverage) with Hubei permutotetra-like virus 9 (Table 2). Despite the blastx result, the genome organization of this virus (Figure 2A) was more similar to Hubei permutotetra-like virus 8 (GenBank accession: NC_033216.1). Both genomes, in fact, contained three coding sequences on the same strand, which simultaneously overlapped for a short region. In our case, the first and the longest ORF, encoding the RdRp, overlapped at the 3′ terminus for 106 nucleotides with both 5′ termini of the other two ORFs, which putatively coded for the capsid proteins. The latter two were found in a +1 and -1 reading frame compared to the RdRp coding sequence. Similarly to all permutotetraviruses, the RdRp motif C was located upstream of motif A, forming a non-canonical C-A-B arrangement. On the basis of these characteristics, hereafter we referred to this virus as Scaphoideus titanus permutotetra-like virus 1. This virus was distantly related to previously described members of the family *Permutotetraviridae* and fell within a weakly supported clade (bootstrap < 70%), which grouped viruses isolated from insects, including the abovementioned Hubei permutotetra-like virus 9 (Figure 1C, Appendix A).

### 3.2. Single-Stranded Negative-Sense RNA Virus

On the basis of the blastx results, three contigs from the St_USA library were identified as the three putative segments of a bunya-like virus, hereafter named Scaphoideus titanus bunya-like virus 1 (Table 2). *Bunyaviridae* have tripartite genomes consisting of a large segment (L), which encodes the RdRp, a medium (M) segment, which encodes the viral glycoprotein precursor, and a small (S) segment, which encodes the nucleoprotein (Appendix A). In the case of Scaphoideus titanus bunya-like virus 1, it was possible to identify the same terminal complementary heptanucleotide in segments M and S, but not in the L segment. Despite that, the RdRp-coding sequence that we putatively assigned to this bunya-like virus was the only one found in this library that showed significant similarities with RdRps of known bunyaviruses. In addition, the phylogenetic analysis unambiguously (100% bootstrap support) placed it within the *Phasmaviridae* cluster, a family of insect-infecting viruses in the order *Bunyavirales* (Figure 3A, Appendix A).

### 3.3. Double-Stranded RNA Viruses

The deduced amino acid sequences of three contigs from St_IT2, St_FR and St_CH libraries were identical and their closest relative identified by blastx (42% amino acid identity, 37% query coverage) was a dsRNA virus named Spissistilus festinus virus 1 (SpFV1) (Table 2). According to the ICTV, less than 50% sequence identity at the protein level generally reflects a species difference, so the newly identified virus was named Scaphoideus titanus toti-like virus 1.

SpFV1 has a monopartite dsRNA genome that shares similarities at the amino acid level with a variety of dsRNA viruses, some unclassified and some classified in the *Totiviridae* or *Chrysoviridae* families [32]. Similarly to SpFV1 and its sister taxon Circulifer tenellus virus 1 (CiTV1) [32], the genome organization of Scaphoideus titanus toti-like virus 1 presented a 5′-leader sequence followed by an ORF encoding a proline-alanine rich protein (PArp), a 3′ proximal RdRp ORF, and finally a short 3′-untranslated region (Figure 2B). The presence of a putative 7nt-frameshift site (AAACCCU) near the PArp ORF stop codon and a downstream stretch of 85 nucleotides that may form a pseudoknot suggested that, similarly to SpFV1 and CiTV1, the expression of the RdRp could be the result of a programmed -1 ribosomal frameshifting (Figure 2B). Such genomic organization is very similar to that of members of the genus *Totivirus*. Despite that, Scaphoideus titanus toti-like virus 1, as well as SpFV1 and CiTV1, fell into a well-supported clade distinct from the one that included viruses belonging to the family *Totiviridae* (Figure 3B, Appendix A). Besides Scaphoideus titanus toti-like virus 1, this small clade included other eight viruses isolated from members of the class Insecta and two viruses identified in plants, Cucurbit yellows-associated virus [33] and Persimmon latent virus [34].

Eleven sequences from the St_USA library were identified as putative segments of a reovirus, hereafter named Scaphoideus titanus reo-like virus 1 (Table 2, Appendix A). Initially, only nine segments were assigned to this virus. The putative missing segments were then manually searched using the genome of Homalodisca vitripennis reovirus (GenBank accessions: FJ497789.1 – FJ497800.1) [35] as bait. The targeted analysis succeeded in finding segments 9 and 11. The fact that both formed chimeric sequences with long insect transcripts was probably the reason for the failure to identify them by automatic blastx analysis. Segment 12 of Homalodisca vitripennis reovirus did not find any significant match with any of the contigs within the St_USA library. The phylogenetic analysis placed Scaphoideus titanus reo-like virus 1 in the phytoreovirus clade, which included Homalodisca vitripennis reovirus and two plant-infecting viruses (Figure 3C, Appendix A). The phytoreovirus clade was, in its turn, phylogenetically close to two reo-like viruses isolated from insects.

In the St_USA library we also identified a partiti-like virus, hereafter named Scaphoideus titanus-associated partiti-like virus 1 (Table 2, Appendix A). A blastx analysis, in fact, revealed the presence of two contigs with similarity to a hypothetical protein (61% amino acid identity, 35% query coverage) and an RdRp (87% amino acid identity, 44% query coverage), both belonging to a partiti-like virus identified in a nematode, named Wuhan large pig roundworm virus 1. The same nine nucleotides were identified at the termini of both segments, confirming they were likely to be part of the same viral genome. Interestingly, Scaphoideus titanus-associated partiti-like virus 1 fell in a large heterogeneous clade that included also deltapartitiviruses (plant-infecting viruses), as well as a group of four viruses identified in arthropods (two from the subphylum Crustacea and two from the class Insecta) and two identified in nematodes (Figure 3D, Appendix A).

### 3.4. Virus Prevalence

RT-PCR was used to define the prevalence of the two viruses discovered in the European libraries. The same RNA extracts used for the construction of RNA-seq libraries were used unpooled to diagnose the presence of Scaphoideus titanus iflavirus 1 and Scaphoideus titanus toti-like virus 1, even in those European samples that, according to the bioinformatic analysis, were negative to one or both viruses (Table 1). We observed a perfect concordance between the presence/absence of the two viruses in the five libraries and the RT-PCR results (Table 1 and Table 2). Regarding the prevalence, results divided by sampling countries are reported in Figure 4. Scaphoideus titanus iflavirus 1 was found only in Italy and in France populations with extremely low frequencies (3% at most). Scaphoideus titanus toti-like virus 1 was the most geographically widespread virus, as it was found in six out of the eight European sampling locations with variable percentages: 24% of the insects sampled in France and 13% of those collected in Switzerland. The percentage of insects negative for both viruses also rather fluctuated, being 100% in samples collected in Hungary and 74% observed in the ones from France.

### 3.5. Endogenous Viral Elements

Some contigs from the European and USA libraries found significant matches with known viruses, but sequences were partial and/or contained multiple stop codons, so they were considered as potential EVEs. This was the case of sequences disclosing similarities to: three fragments of RdRps from viruses belonging to the order *Mononegavirales*, a glycoprotein from a phlebovirus (order *Bunyavirales*), an RdRp from a reovirus, multiple non-structural proteins (NS1) and capsid proteins (VP1) from members of the family *Parvoviridae*.

Given that *S. titanus* genome is not available, specific primers were designed on one sequence for each phylogenetic group that is characterized by an RNA genome without DNA intermediates, i.e., mononegaviruses, reoviruses and bunyaviruses (Appendix A). PCR analyses on total DNA extracted from *S. titanus* revealed that they were indeed all endogenised into the host genome (data not shown). The potential integration of sequences similar to parvovirus genomes could not be verified by PCR, as they are DNA viruses and a PCR amplification could be due to either, an integration into the host genome or simply to the presence of the virus within the insect.

Additionally, multiple fragments with similarities to the PArp from Spissistilus festinus virus 1 were found in each European library, but, unlike the EVEs mentioned above, they were characterized by ORFs that spanned the whole sequence without any premature stop codon. Notably, none of these sequences could be assigned to the previously described genome of Scaphoideus titanus toti-like virus 1. It was possible to reconstruct a large part of the 3′ PArp coding sequence only by using transcripts from all the European libraries together, but still the deduced protein was partial. In addition, a specific search to identify the missing RdRp failed. The fragmentation of the PArp coding sequence observed in each library and the lack of the corresponding RdRp represented clues in favor of a possible integration of these sequences into the insect genome. Specific primers designed on one of these fragments (Appendix A), which was found in two out of the five European libraries, revealed that at least this part of the viral genome was indeed integrated into the insect genome (data not shown).

## 4. Discussion

Pioneered by studies of bacteriophages in the marine environment [36], undirected high-throughput sequencing is becoming the easiest and cheapest procedure for the detection and genetic characterization of new viruses or virus variants. Metagenomic and metatranscriptomic studies have been radically changing virology, re-shaping virus lineages and revealing an unexpected richness of genome sizes and structures [37]. Although such astonishingly abundant sequences are linked to little or no biological information, there is unanimity in the assumption that genome sequences can be used to support the existence of a free/replicating virus [38]. In our work we chose a metatranscriptomic approach to characterize the virome of *S. titanus*, primarily because arthropods demonstrated to harbor an unprecedented diversity of RNA viruses [28]. Secondly, transcriptome profiling can provide additional information, such as the presence of RNAs possibly expressed by DNA viruses as well as a rough quantification of the viral load within the host.

Virus-like sequences often derive from mixed virus populations and, consequently, there is the risk of assembling chimeric or artefactual genomes, neglecting sequences from viruses with segmented and multipartite genomes, and assigning transcribed virus-derived sequences integrated into the host genome to actively replicating viruses [37]. Stringent criteria were adopted in this work to minimize, as far as possible, the bioinformatic pitfalls that may lead to mis-assemblies and misinterpretation of the newly discovered genomes. First, we only considered contigs that encoded complete, or near complete, viral genomes with intact open reading frames to avoid EVEs. Additionally, in the case of multipartite viruses, we looked for the presence of the same nucleotides at the termini of each segment to ensure the correct genome assignment. The low and variable prevalence of the two newly identified viruses in the European populations of *S. titanus* constituted a further clue in favor of true replicating viruses. On the opposite, the presence in all or almost all libraries of some viral sequences with premature stop codons supported the hypothesis that they were potential EVEs. Some of these were unambiguously demonstrated to be integrated into *S. titanus* genome and expressed as RNAs.

Both metatranscriptomics and metagenomics rely solely on inferred homology to known viruses to identify putative viral sequences, as viruses do not share any specific genetic marker. Such an approach, however, precludes the discovery of viruses that lack closely related sequences. Consequently, we could not exclude that our libraries contained more viruses than the nine described in this work. Another issue raised by such sequencing studies is whether the newly discovered viruses are true infectious agents of the targeted host. Field populations of insects can, in fact, harbour viruses that infect the insect itself, viruses of microbes or pathogens associated with the insect, and viruses associated with ingested materials [39]. Viruses infecting insect microflora, in particular, could also be potential candidates for insect biocontrol (e.g., bacteriophages attacking primary bacteria symbionts), but their exploitation would require the knowledge of the exact composition of *S. titanus* microbioma, which is so far fairly unexplored. All samples used in this work were caught in the wild, so the chance that some of the sequenced microorganisms were associated with insects only as environmental contaminants was very high. Therefore, we preferred to exclude bacteriophages and known plant viruses (e.g., Grapevine asteroid mosaic associated virus found in St_CH library) and retained only those viruses that are closely related to other insect-infecting viruses. The main ambiguity was represented by Scaphoideus titanus-associated partiti-like virus 1. After the taxonomic reorganization of *Partitiviridae*, the family has now five genera: *Alphapartitivirus* and *Betapartitivirus* (fungi and plant viruses), *Gammapartitivirus* (fungi viruses) *Deltapartitivirus* (plant viruses) and *Cryspovirus* (protozoa viruses) [40]. Scaphoideus titanus-associated partiti-like virus 1 clustered with two viruses identified in nematodes and formed a distinct clade only distantly related to *Deltapartitivirus,* the phylogenetically closest genus among those officially recognized by the ICTV. This result was consistent with phylogenetic analyses presented in previous works [41,42], especially in the manuscript by Shi et al. [28], which remarked that in the *Partitiviridae* family it is possible to resolve well-supported clades that appear to be specific to invertebrate hosts. Despite that, conclusive evidence that a partiti-like virus could infect insects has never been reported in literature, so we preferred the indication “associated virus”.

*Scaphoideus titanus* is native to North America and invasive in Europe. Sampling in Europe was quite straightforward, because this species is abundant in vineyards showing symptoms of the Flavescence dorée disease. On the contrary, in the USA, the presence of *S. titanus* is not related to the manifestation of specific symptoms in the vegetation. Sampling in the USA is further complicated by the fact that *S. titanus* is usually found in unmanaged open woodlands and in mixed populations with other morphologically similar species, making its correct identification quite problematic, while sampling [43]. Despite the low number of insects caught in the USA, the most striking result of this investigation was the marked difference in the diversity and abundance of the viral species identified in the two geographical areas. Seven viral species were discovered from three individuals collected in the USA, whereas only two viral species, Scaphoideus titanus iflavirus 1 and Scaphoideus titanus toti-like virus 1, were collectively identified in the 214 individuals that were analyzed from various European sites. Additionally, 91% of the insects collected throughout Europe appeared to be free from both viruses. The Enemy Release Hypothesis (ERH) could be a possible explanation of such a dramatic shift in the number of viruses putatively infecting *S. titanus.* ERH predicts that a species will be successful in a new habitat as a result of a drop in overall number of natural enemies [44], e.g., predators, parasites and pathogens, including viruses. This hypothesis has been widely applied to invasive species, such as plant or animal pests in new habitats, even if the outcomes of these studies are highly variable and sometimes even contradictory [45]. A similar change in the profile of insect viruses associated with invasion of new habitats was also observed in *Drosophila suzukii*, a dipteran pest of soft fruits first isolated in Japan at the beginning of the 20^th^ century and now spread worldwide [46].

Although the inventory of viruses identified in *S. titanus* does not include any member of the family *Baculoviridae,* the species identified in this work have the potential to be used as natural enemies to control the insect vector. Unfortunately, most of them were identified in the St_USA library only and will not be given priority as potential biocontrol agents for the European *S. titanus* populations, in order to avoid issues related to the introduction of alien viral species. As far as it concerns the remaining two species, Scaphoideus titanus iflavirus 1 and Scaphoideus titanus toti-like virus 1, experimental investigations, such as isolation and pathogenicity assays, are required before proposing them as components of integrated pest management strategies. Viruses may not be necessarily lethal to their host, but just cause a reduction in its survival and fecundity, which in our case may decrease the overall insect density in vineyards and thereby Flavescence dorée incidence. Besides targeting insect vectors directly, they might also interfere with FDp acquisition and transmission. Studies on *Aedes aegypti*, in fact, demonstrated the existence of reciprocal interactions among insect antiviral, anti-bacterial, and anti-parasite immune responses that involve the Toll immune signaling pathway [47,48].

In addition to wild-type viruses, biotechnologically manipulated viruses in the form of VIGS could be used to interfere with the insect immune system or manipulate the expression of insect genes that are hypothesized to be associated with the acquisition and transmission mechanisms of phytoplasmas. From this perspective, Scaphoideus titanus iflavirus 1 could be considered a good candidate for the construction of infectious clones. Indeed, an appropriate strategy to deliver virus-based biocontrol agents should be developed, especially for piercing-sucking insects like *S. titanus*. In-field application of RNA silencing techniques in phloem-feeding insects poses, in fact, a great technical challenge, since the delivery should preferably occur through vascular tissues. In this regard, double-stranded RNAs (dsRNAs) have been successfully delivered to psyllids and leafhoppers by root drench and trunk injections of grapevine and Citrus plants [49]. Moreover, topical application of dsRNAs has been used as an alternative way to induce gene silencing in the Asian citrus psyllid, *Diaphorina citri* [50]. Both approaches might also be explored for VIGS delivery.

The majority of meta-omics studies about the virosphere of insect vectors has been conducted mostly on species transmitting diseases of medical and veterinary importance, such as *Anopheles* [51,52]*, Aedes* [53,54] and *Culex* [42,55]. By contrast, investigations on viruses that may infect insect vectors of plant pathogens are scarce. Metagenomics study of *Bemisia tabaci,* vector of several plant viruses, resulted in the identification of many RNA viruses, some of which are likely to infect the whiteflies themselves [56]. New putative viruses that may have the potential to be used as biocontrol agents have been discovered in *D. citri*, the natural vector of the causal agent of Huanglongbing (HLB) [57]. Novel dicistrovirus-like sequences were also identified during a metagenomics project of another vector for plant-infecting viruses, *Aphis fabae* [58]. The nine new viral genomes described in this work probably do not depict the total diversity of viruses that may infect *S. titanus*, as it is known that the composition of insect viromes is influenced by a variety of factors, such as diet, stage of development, geographical location [59,60]. Yet, we provide first insights into the unexplored viral community of this vector, which is responsible for the spread of FDp in European vineyards. The possibility of exploiting these naturally occurring viruses, as specific and targeted biocontrol agents of *S. titanus,* would be a possible answer to the increasing demand for a more sustainable viticulture.

## Figures and Tables

**Figure 1 viruses-12-00287-f001:**
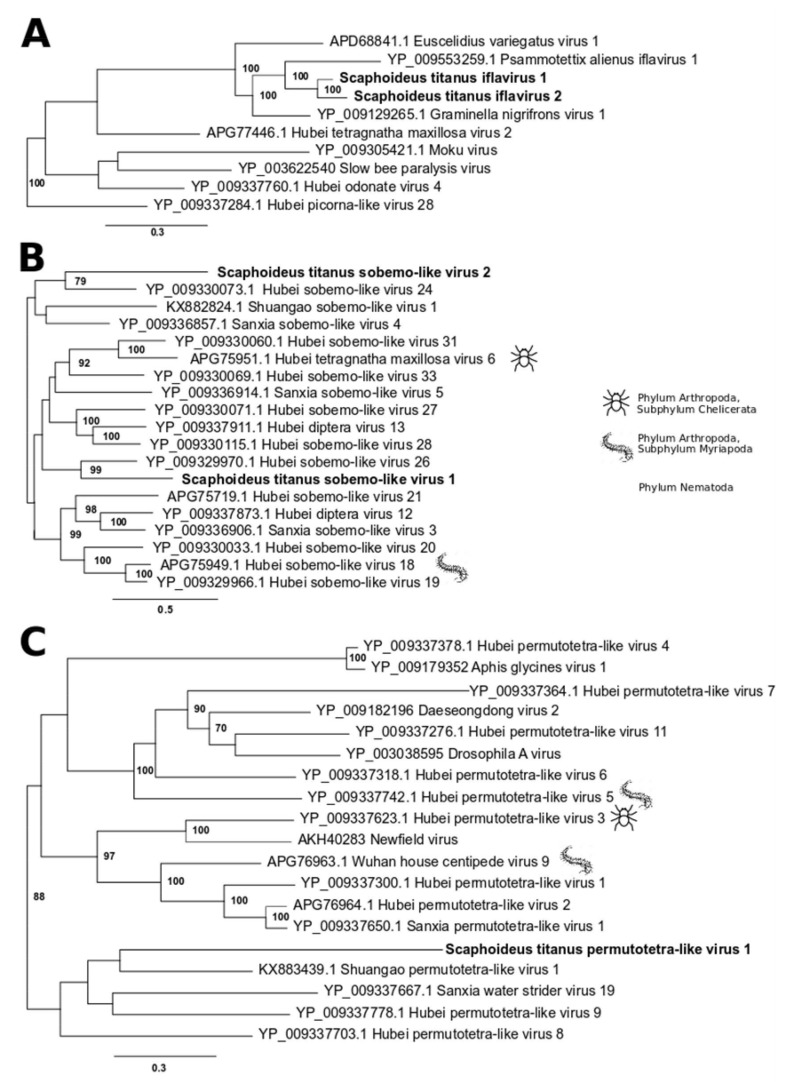
Maximum likelihood phylogeny of +ssRNA viruses. Putative viruses that may infect *S. titanus* are in bold. All the phylogenies described in this manuscript used virus sequences and nomenclature as provided by Shi et al. [28]. Tree (**A**): Scaphoideus titanus iflavirus 1 and Scaphoideus titanus iflavirus 2 within the “Iflaviridae-Secoviridae related cluster”. GenBank accession numbers of the viruses included in the phylogenetic analysis are as follows: Euscelidius variegatus virus 1, APD68841.1; Psammotettix alienus iflavirus 1, YP_009553259.1; Graminella nigrifrons virus 1, YP_009129265.1; Hubei tetragnatha maxillosa virus 2, APG77446.1; Moku virus, YP_009305421.1; Slow bee paralysis virus, YP_003622540; Hubei odonate virus 4, YP_009337760.1; Hubei picorna-like virus 28, YP_009337284.1. Tree (**B**): Scaphoideus titanus sobemo-like virus 1 and Scaphoideus titanus sobemo-like virus 2 within the “Luteo-Sobemo” clade. GenBank accession numbers of the viruses included in the phylogenetic analysis are as follows: Hubei sobemo-like virus 24, YP_009330073.1; Shuangao sobemo-like virus 1, KX882824.1; Sanxia sobemo-like virus 4, YP_009336857.1; Hubei sobemo-like virus 31, YP_009330060.1; Hubei tetragnatha maxillosa virus 6, APG75951.1; Hubei sobemo-like virus 33, YP_009330069.1; Sanxia sobemo-like virus 5, YP_009336914.1; Hubei sobemo-like virus 27, YP_009330071.1; Hubei diptera virus 13, YP_009337911.1; Hubei sobemo-like virus 28, YP_009330115.1; Hubei sobemo-like virus 26, YP_009329970.1; Hubei sobemo-like virus 21, APG75719.1; Hubei diptera virus 12, YP_009337873.1; Sanxia sobemo-like virus 3, YP_009336906.1; Hubei sobemo-like virus 20, YP_009330033.1; Hubei sobemo-like virus 18, APG75949.1; Hubei sobemo-like virus 19, YP_009329966.1. Tree (**C**): Scaphoideus titanus permutotetra-like virus 1 within the “Permutotetra” clade. GenBank accession numbers of the viruses included in the phylogenetic analysis are as follows: Hubei permutotetra-like virus 4, YP_009337378.1; Aphis glycines virus 1, YP_009179352_; Hubei permutotetra-like virus 7, YP_009337364.1; Daeseongdong virus 2, YP_009182196; Hubei_permutotetra-like virus 11, YP_009337276.1; Drosophila A virus, YP_003038595; Hubei permutotetra-like virus 6, YP_009337318.1; Hubei permutotetra-like virus 5, YP_009337742.1; Hubei permutotetra-like virus 3, YP_009337623.1; Newfield virus, AKH40283; Wuhan house centipede virus 9, APG76963.1; Hubei permutotetra-like virus 1, YP_009337300.1; Hubei permutotetra-like virus 2, APG76964.1; Sanxia_permutotetra-like virus 1, YP_009337650.1; Shuangao permutotetra-like virus 1, KX883439.1; Sanxia water strider virus 19, YP_009337667.1; Hubei permutotetra-like virus 9, YP_009337778.1; Hubei permutotetra-like virus 8, YP_009337703.1.Only bootstrap values higher than 70 are shown. Complete trees are provided in Appendix A. All viruses included in this figure were isolated from members of the class Insecta, with the exceptions of those with silhouetted animals on the right. Silhouettes were downloaded from http://www.iconarchive.com/. The scale bars indicate the evolutionary distance expressed as amino acid substitutions per site.

**Figure 2 viruses-12-00287-f002:**
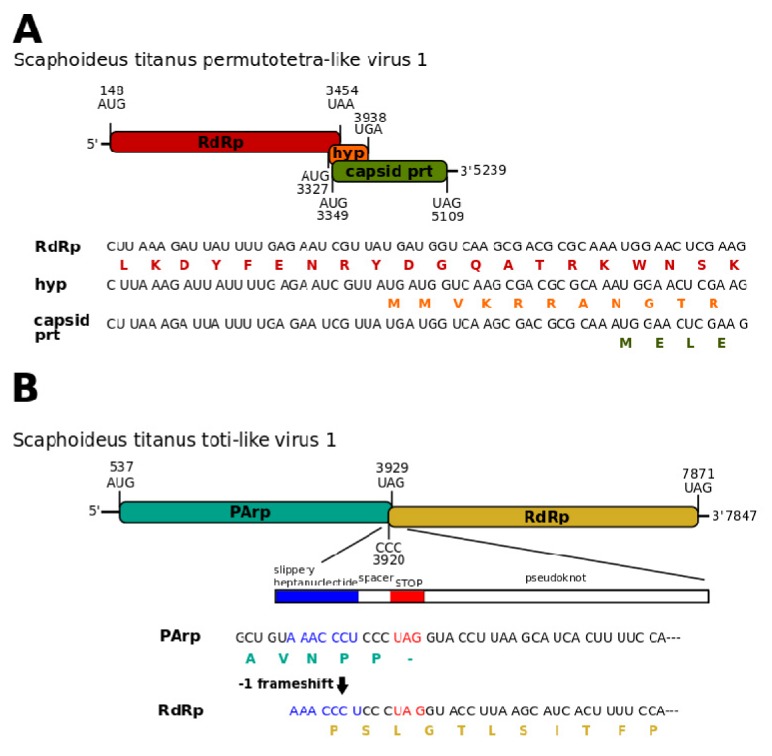
Structures of Scaphoideus titanus permutotetra-like virus 1 and Scaphoideus titanus toti-like virus 1. (**A**) The organization of the genome of Scaphoideus titanus permutotetra-like virus 1; (**B**) the organization of the genomes of Scaphoideus titanus toti-like virus 1 found in the St_IT2, St_FR and St_CH libraries is the same, but the 5′ and 3′ UTRs show different lengths. In this figure we specifically illustrated the genome assembled from the St_IT2 library. Boxes represent boundaries of ORFs. RdRp: RNA-dependent RNA polymerase. Hyp: hypothetical protein. Capsid prt: capsid protein. PArp: proline-alanine rich protein.

**Figure 3 viruses-12-00287-f003:**
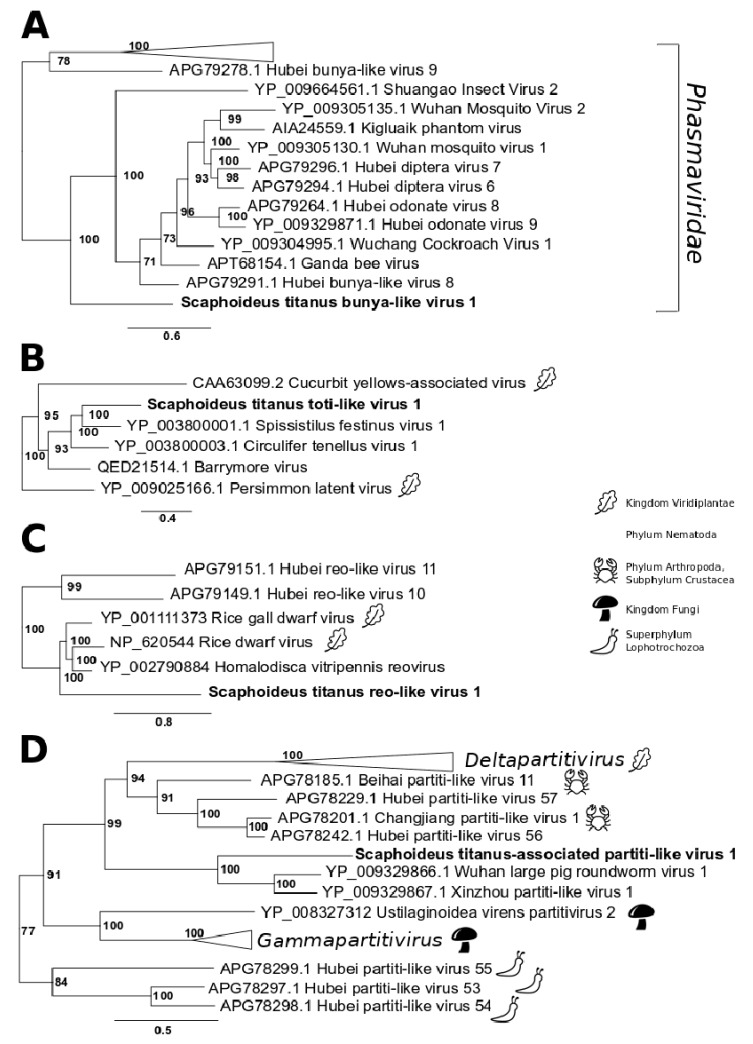
Maximum likelihood phylogeny of -ssRNA and dsRNA viruses. Putative viruses that may infect *S. titanus* are in bold. All the phylogenies described in this manuscript used virus sequences and nomenclature as provided by Shi et al. [28]. Tree (**A**): Scaphoideus titanus bunya-like virus 1 within the “Bunya-Arena” clade; GenBank accession numbers of the viruses included in the phylogenetic analysis are as follows: Hubei bunya-like virus 9, APG79278.1; Shuangao Insect Virus 2, YP_009664561.1; Wuhan Mosquito Virus 2, YP_009305135.1; Kigluaik phantom virus, AIA24559.1; Wuhan mosquito virus 1, YP_009305130.1; Hubei diptera virus 7, APG79296.1; Hubei diptera virus 6, APG79294.1; Hubei odonate virus 8, APG79264.1; Hubei odonate virus 9, YP_009329871.1; Wuchang Cockroach Virus 1, YP_009304995.1; Ganda bee virus, APT68154.1; Hubei bunya-like virus, 8 APG79291.1. The collapsed branched named “*Phasmaviridae”* grouped five sequences: Jonchet virus, AKN56871; Hubei bunya-like virus 10, APG79271.1; Wuhan Insect virus 2, YP_009270651.1; Sanxia Water Strider Virus 2, AJG39245.1; Ferak virus, AKN56888. Tree (**B**): Scaphoideus titanus toti-like virus 1 within the “Toti-Chryso” clade. GenBank accession numbers of the viruses included in the phylogenetic analysis are as follows: Cucurbit yellows-associated virus, CAA63099.2; Spissistilus festinus virus 1, YP_003800001.1; Circulifer tenellus virus 1, YP_003800003.1; Barrymore virus, QED21514.1; Persimmon latent virus, YP_009025166.1. Tree (**C**): Scaphoideus titanus reo-like virus 1 within the “Reo” clade. GenBank accession numbers of the viruses included in the phylogenetic analysis are as follows: Hubei reo-like virus 11, APG79151.1; Hubei reo-like virus 10, APG79149.1; Rice gall dwarf virus, YP_001111373; Rice dwarf virus, NP_620544; Homalodisca vitripennis reovirus, YP_002790884. Tree (**D**): Scaphoideus titanus associated partiti-like virus 1 within the “Partiti-Picobirna” clade. GenBank accession numbers of the viruses included in the phylogenetic analysis are as follows: Beihai partiti-like virus 11, APG78185.1; Hubei partiti-like virus 57, APG78229.1; Changjiang partiti-like virus 1, APG78201.1; Hubei partiti-like virus 56, APG78242.1; Wuhan large pig roundworm virus 1, YP_009329866.1; Xinzhou partiti-like virus 1, YP_009329867.1; Ustilaginoidea virens partitivirus 2, YP_008327312; Hubei partiti-like virus 55, APG78299.1; Hubei partiti-like virus 53, APG78297.1; Hubei partiti-like virus 54, APG78298.1. The collapsed branched named “*Deltapartitivirus”* grouped six sequences: Fig cryptic virus, YP_004429258; Rose cryptic virus 1, YP_001686786; Raphanus sativus cryptic virus 2, YP_001686783; Raphanus sativus cryptic virus 3, YP_002364401; Hubei partiti-like virus 58, APG78223.1; Persimmon cryptic virus, YP_006390091. The collapsed branched named “*Gammapartitivirus”* grouped five sequences: Fusarium solani virus 1, NP_624350.1; Penicillium stoloniferum virus S, YP_052856.2; Gremmeniella abietina RNA virus MS1, NP_659027.1; Botryotinia fuckeliana partitivirus 1, YP_001686789.1; Discula destructiva virus 1, NP_116716.1. Only bootstrap values higher than 70 are shown. Complete trees are provided in Appendix A.All viruses included in this figure were isolated from members of the class Insecta, with the exceptions of those with silhouetted animals, fungi or plants on the right. Silhouettes were downloaded from http://www.iconarchive.com/. The scale bars indicate the evolutionary distance expressed as amino acid substitutions per site.

**Figure 4 viruses-12-00287-f004:**
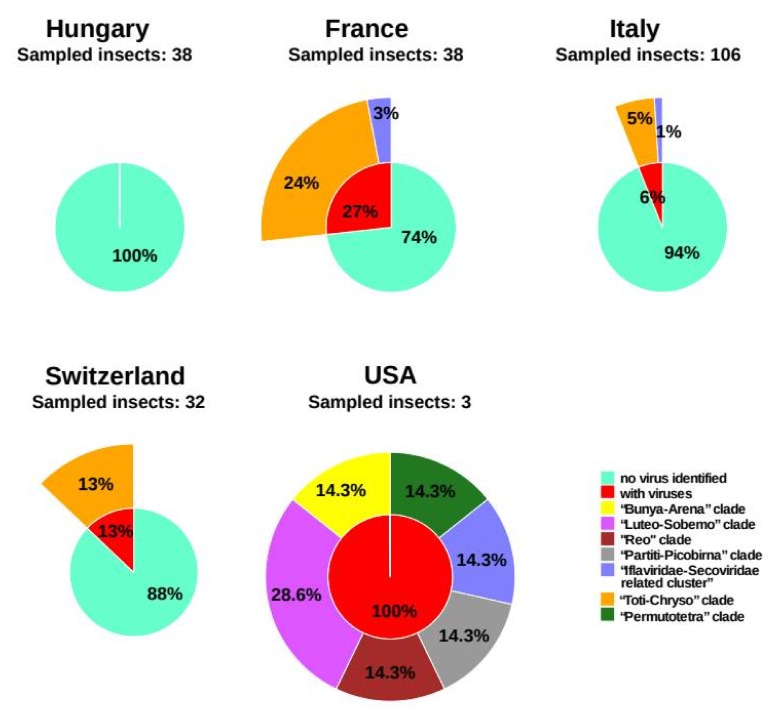
Virus clades and clusters identified in Europe and in the USA. Each sunburst reports prevalence of the newly identified virus in the 5 sampling countries and assignment to viral clades and clusters, according to Shi et al. [28] (see also Table 1). Sunbursts were generated by R (version 3.4.4) packages “cowplot” and “tidyverse”.

**Table 1 viruses-12-00287-t001:** Sampling information, RNA-seq pooling strategies and prevalence of Scaphoideus titanus iflavirus 1 and Scaphoideus titanus toti-like virus 1 in the European *S. titanus* populations.

RNA-seq LibrariesID	**Country**	State/Region	Sampling Sites	Latitude (Decimal Degrees)	Longitude (Decimal Degrees)	Sampling Date (yyyy/mm/dd)	Number of Sampled Individuals	Positive to Scaphoideus titanus Iflavirus 1	Positive to Scaphoideus titanustoti-Like Virus 1	
St_HU	Hungary	Pest	Monor	47.354861	19.472177	2018-07-12	20	0	0	
St_HU	Hungary	Pest	Gomba	47.349622	19.505863	2018-07-12	18	0	0	
St_FR	France	Burgundy	Corgoloin	47.095667	4.908233	2018-07-06	20	0	8	
St_FR	France	Dordogne	Saint-Nexans	44.472132	0.345279	2018-07-10	18	1	1	
St_IT1	Italy	Piedmont	Asti	44.921854	8.195758	2018-07-31	4	0	0	
St_IT1	Italy	Piedmont	Montà	44.807434	7.965787	2018-07-31	12	1	0	
St_IT1	Italy	Piedmont	Portacomaro	44.962898	8.26072	2018-08-03	16	0	0	
St_IT1	Italy	Piedmont	Cisterna	44.82591	8.008371	2018-08-03	4	0	0	
St_IT1	Italy	Lombardy	Ome	45.6302972	10.129867	2018-07-11	4	0	0	
St_IT1	Italy	Veneto	Verona	45.4227833	11.025431	2018-08-08	8	0	0	
St_IT2	Italy	Friuli-Venezia Giulia	Togliano	46.1129556	13.411311	2018-07-30	22	0	2	
St_IT2	Italy	Friuli-Venezia Giulia	Savorgnano al Torre	46.161786	13.286864	2018-07-31	24	0	1	
St_IT2	Italy	Abruzzo	Vacri	42.290333	14.223861	2018-07-16	12	0	2	
St_CH	Switzerland	Canton Vaud	Gland	46.42616	6.28217	2018-08-09	16	0	4	
St_CH	Switzerland	Canton Valais	Pramagnon	46.24987	7.45371	2018-08-09	16	0	0	
							**214**	**2**	**18**	**Total**
St_USA	USA	Illinois	Muncie	40.0893	-87.8323	2018-06-11	3			
**Insects for DNA extraction^1^**	**Country**	**State/Region**	**Sampling Sites**	**Latitude (Decimal Degrees)**	**Longitude (Decimal Degrees)**	**Sampling Date** **(yyyy/mm/dd)**	**Number of Sampled Individuals**			
	Italy	Piedmont	Montà	44.807434	7.965787	2018-07-31	2			
	Italy	Piedmont	Dogliani	44.5362837	7.9312913	2018-08-20	2			

^1^ The second section of the table reports sampling information about insects used to demonstrate the integration of some viral sequences into the insect genome.

**Table 2 viruses-12-00287-t002:** List of the sequences assigned to putative insect-infecting viruses found in each RNA-seq library. Column 9 reports the kind (+ssRNA, -ssRNA and dsRNA) and structure (monopartite/multipartite) of the viral genomes identified in this work. The identification of segments in some multipartite genomes is also indicated.

Library ID	Accession Numbers	Sequence Length (bp)	Best Blastx Hit Description	% Identity	% Query Coverage	RPKM	Tentative Name	Genome
St_IT1	MN982380	10729	YP_009129265.1 polyprotein [Graminella nigrifrons virus 1]	51	83	0.01	Scaphoideus titanus iflavirus 1	+ssRNA (monopartite)
St_IT2	MN982382	7847	YP_003800001.1 RNA-directed RNA polymerase partial [Spissistilus festinus virus 1]	42	37	0.002	Scaphoideus titanus toti-like virus 1	dsRNA (monopartite)
St_CH	MN982383	7839	YP_003800001.1 RNA-directed RNA polymerase partial [Spissistilus festinus virus 1]	42	37	0.032	Scaphoideus titanus toti-like virus 1	dsRNA (monopartite)
St_FR	MN982379	10693	YP_009129265.1 polyprotein [Graminella nigrifrons virus 1]	51	83	0.004	Scaphoideus titanus iflavirus 1	+ssRNA (monopartite)
St_FR	MN982381	7871	YP_003800001.1 RNA-directed RNA polymerase partial [Spissistilus festinus virus 1]	42	37	0.023	Scaphoideus titanus toti-like virus 1	dsRNA (monopartite)
St_USA	MN982384	4361	API65464.1 glycoprotein precursor [Sanxia Water Strider Virus 2]	25	12	1.016	Scaphoideus titanus bunya-like virus 1	−ssRNA (tripartite): segment M
St_USA	MN982385	1458	APG79297.1 putative nucleoprotein [Hubei diptera virus 7]	27	43	0.9	Scaphoideus titanus bunya-like virus 1	−ssRNA (tripartite): segment S
St_USA	MN982386	7740	APT68154.1 RNA-dependent RNA polymerase [Ganda bee virus]	33	74	0.359	Scaphoideus titanus bunya-like virus 1	−ssRNA (tripartite): segment L
St_USA	MN982387	10519	YP_009129265.1 polyprotein [Graminella nigrifrons virus 1]	51	85	0.045	Scaphoideus titanus iflavirus 2	+ssRNA (monopartite)
St_USA	MN982388	1455	AWA82259.1 hypothetical protein, partial [Naganuma virus]	38	91	0.145	Scaphoideus titanus sobemo-like virus 1	+ssRNA (bipartite)
St_USA	MN982389	3213	YP_009329970.1 hypothetical protein [Hubei sobemo-like virus 26]	60	99	0.103	Scaphoideus titanus sobemo-like virus 1	+ssRNA (bipartite)
St_USA	MN982390	1550	AWA82259.1 hypothetical protein partial [Naganuma virus]	36	77	0.231	Scaphoideus titanus sobemo-like virus 2	+ssRNA (bipartite)
St_USA	MN982391	3100	YP_009330073.1 hypothetical protein 2 [Hubei sobemo-like virus 24]	46	97	0.172	Scaphoideus titanus sobemo-like virus 2	+ssRNA (bipartite)
St_USA	MN982393	1303	YP_009329884.1 hypothetical protein [Wuhan large pig roundworm virus 1]	35	61	0.227	Scaphoideus titanus-associated partiti-like virus 1	dsRNA (bipartite)
St_USA	MN982392	1558	AVV63192.1 RNA-directed RNA polymerase [Wuhan large pig roundworm virus]	44	87	0.272	Scaphoideus titanus-associated partiti-like virus 1	dsRNA (bipartite)
St_USA	MN982394	5239	YP_009337778.1 RdRp [Hubei permutotetra-like virus 9]	39	39	0.764	Scaphoideus titanus permutotetra-like virus 1	+ssRNA (monopartite)
St_USA	MN982397	1240	ADN64742.1 non-structural protein [Homalodisca vitripennis reo-like virus]	79	58	2.378	Scaphoideus titanus reo-like virus 1	dsRNA (multipartite): segment 10
St_USA	MN982398	1867	ADN64768.1 minor core protein [Homalodisca vitripennis reo-like virus]	83	42	1.146	Scaphoideus titanus reo-like virus 1	dsRNA (multipartite): segment 7
St_USA	MN982399	2937	ADN64783.1 minor core protein [Homalodisca vitripennis reo-like virus]	52	49	2.661	Scaphoideus titanus reo-like virus 1	dsRNA (multipartite): segment 5
St_USA	MN982395	4957	ADN64816.1 RNA-directed RNA polymerase [Homalodisca vitripennis reo-like virus]	87	61	5.053	Scaphoideus titanus reo-like virus 1	dsRNA (multipartite): segment 1
St_USA	MN982400	3560	YP_002790885.1 RNA-binding protein [Homalodisca vitripennis reo-like virus]	96	49	5.444	Scaphoideus titanus reo-like virus 1	dsRNA (multipartite): segment 2
St_USA	MN982401	1416	Q85451.1 RecName: Full=Outer capsid protein P8	62	88	1.679	Scaphoideus titanus reo-like virus 1	dsRNA (multipartite): segment 8
St_USA	MN982402	3179	ADN64799.1 major core protein [Homalodisca vitripennis reo-like virus]	95	55	1.901	Scaphoideus titanus reo-like virus 1	dsRNA (multipartite): segment 3
St_USA	MN982403	2648	YP_002790887.1 zinc-finger protein [Homalodisca vitripennis reo-like virus]	73	30	3.571	Scaphoideus titanus reo-like virus 1	dsRNA (multipartite): segment 4
St_USA	MN982404	1778	YP_002790889.1 non-structural protein [Homalodisca vitripennis reo-like virus]	78	33	1.196	Scaphoideus titanus reo-like virus 1	dsRNA (multipartite): segment 6
St_USA	MN982396	1179	YP_002790892.1 non-structural protein [Homalodisca vitripennis reo-like virus]	68	47	1.802	Scaphoideus titanus reo-like virus 1	dsRNA (multipartite): segment 9
St_USA	MN982405	798	YP_002790894.1 non-structural protein [Homalodisca vitripennis reo-like virus]	48	45	4.192	Scaphoideus titanus reo-like virus 1	dsRNA (multipartite): segment 11

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
