# Peer review of "New Viral Sequences Identified in the Flavescence Dorée Phytoplasma Vector *Scaphoideus titanus"

_viruses, 2020, doi:10.3390/v12030287_

Round 1

Reviewer 1 Report

The objective of the study was to identify the viruses present in the wild populations of Neartic leafhopper species a vector of phytoplasma. A metatranscriptomic approach was taken to identify the viruses present in the adult hoppers collected from Europe and USA. The study identified new viruses in the four samples collected in Europe and one sample collected in USA.

The analysis was thoroughly performed to identify the viral genomes. Explanation provided in the results and the discussion is well written and are cohesive.

There are some minor revisions that the authors need to address in the manuscript particularly in methods section and also figures. More details regarding the package versions and parameters used for bioinformatic analysis need to be included. Please provide the alignment figures for all the viruses identified in the supplementary or as a concise and composite figure in the main manuscript to visualize the alignments done to identify the viral genomes (Figures similar to Figure 2).

Please find the specific comments below

Line 34: Include if the neartic leafhopper is a phloem or xylem feeder

Line 88: Why were only few samples collected from USA?

Line 50: Rephrase the line to make it simple

Line 113: Was the manufacturer’s protocol followed as is?

Line 116: Please mention the versions used, parameters used and also the source of the databases downloaded.

Line 121: Why were the house, human and dog sequences were searched for within the library?

Line 127: The rationale for removing the viruses present in insect diet, microflora is not strong as these viruses could also be potential candidates for application.

Line 168: Which program was used to perform Blast to NCBI database?

Line 275: Include the sequences used to build the phylogenetic tree in the figure legend for all the figures.

Line 532: Please explain the relationship between the enemy release hypothesis and not being able to find viruses in the Europe samples. Are viruses considered the enemies?

Reviewer 2 Report

Dear authors,

I have read with interest your MS about novel viruses recognized in Flavescens dorée vector Scaphoideus titanus. I have to say that this work was well done and clearly written.

I have no comments or suggestions on methods and results. Complex phylogenetic trees are placed in Supplementary, more suited trees in condensed form are presented in text.

After correction of typing errors (line 98 - should be "spectrophotometer", line 575 - should be "metagenomics") and errors in References (typography of journal´s name in references 2, 36, 41, 49, 50, 55), errors in author´s names in references 11, 38, deleting "others" in references 3 and 11, and providing more details (chapter, pages) of reference 31, the MS should be published as is. 

Author Response

After correction of typing errors (line 98 - should be "spectrophotometer", line 575 - should be "metagenomics") and errors in References (typography of journal´s name in references 2, 36, 41, 49, 50, 55), errors in author´s names in references 11, 38, deleting "others" in references 3 and 11, and providing more details (chapter, pages) of reference 31, the MS should be published as is.

All errors have been corrected.